# A Review on Coupled Bulk Acoustic Wave MEMS Resonators

**DOI:** 10.3390/s22103857

**Published:** 2022-05-19

**Authors:** Linlin Wang, Chen Wang, Yuan Wang, Aojie Quan, Masoumeh Keshavarz, Bernardo Pereira Madeira, Hemin Zhang, Chenxi Wang, Michael Kraft

**Affiliations:** 1Micro- and Nanosystems—MNS, Department of Electrical Engineering ESAT, KU Leuven, B-3001 Leuven, Belgium; linlin.wang@kuleuven.be (L.W.); aojie.quan@kuleuven.be (A.Q.); masoumeh.keshavarz@kuleuven.be (M.K.); bernardo.pereiramadeira@kuleuven.be (B.P.M.); hemin.zhang@kuleuven.be (H.Z.); chenxi.wang@kuleuven.be (C.W.); michael.kraft@kuleuven.be (M.K.); 2Department of Electrical and Electronic Engineering, Huazhong University of Science and Technology, Wuhan 430074, China

**Keywords:** Micro-Electro-Mechanical System (MEMS), bulk acoustic wave (BAW), coupled resonators, sensing theory, transduction mechanism

## Abstract

With the introduction of the working principle of coupled resonators, the coupled bulk acoustic wave (BAW) Micro-Electro-Mechanical System (MEMS) resonators have been attracting much attention. In this paper, coupled BAW MEMS resonators are discussed, including the coupling theory, the actuation and sensing theory, the transduction mechanism, and the applications. BAW MEMS resonators normally exhibit two types of vibration modes: lateral (in-plane) modes and flexural (out-of-plane) modes. Compared to flexural modes, lateral modes exhibit a higher stiffness with a higher operating frequency, resulting in a lower internal loss. Also, the lateral mode has a higher Q factor, as the fluid damping imposes less influence on the in-plane motion. The coupled BAW MEMS resonators in these two vibration modes are investigated in this work and their applications for sensing, timing, and frequency reference are also presented.

## 1. Introduction

Micro-Electro-Mechanical System (MEMS) is a research field that uses mechanical structures often fabricated in the bulk of a wafer [1]. Making use of the concept of relating their mechanical and electrical components enables them to be capable of sense, control and actuate on the micro-scale. A sub-class that has recently received increasing attention and has become one of the most important building blocks of MEMS devices, is micromachined resonant devices, i.e., their mechanically resonating micro-structures are electrically brought into resonance [2,3].

Among all resonant devices, a specific type of MEMS resonator is the acoustic wave resonator, which is the device exploiting acoustic wave propagation and is naturally vibrating at a resonance frequency related to their dimension and mechanical material properties [4]. There are two main types of acoustic wave resonators: surface acoustic wave (SAW) resonators [4,5,6], in which the propagation of acoustic waves is on the surface of the substrate, and bulk acoustic wave (BAW) resonators [4,7,8], where the acoustic waves propagate through the bulk of the substrate. BAW resonators normally feature high bulk moduli [9], high operating frequencies, as well as high quality (Q) factors [10] and thus are widely used as filters [3,11,12,13,14,15], oscillators [3,13,16,17,18] as well as sensors [3,15,19,20,21,22].

In terms of the ways of acoustic wave propagation, BAW resonators mainly have two kinds of modes, i.e., out-of-plane [23] and in-plane modes [24,25]. Out-of-plane modes, also named flexural modes [4], include the longitudinal [4] and anti-symmetric lamb modes [25]. In-plane modes, also termed lateral modes [4,26], include transverse/shear [4] and symmetric lamb modes [25]. Compared to BAW resonators with flexural modes, those with lateral vibration modes can exhibit a higher stiffness, resulting in a higher resonance frequency. Since the relatively high stiffness of the lateral modes typically let a BAW resonator undergo small deformations, the internal mechanical loss, such as thermoelastic damping (TED) [27], can be reduced [28]. Moreover, a BAW resonator working in the lateral vibration mode has only in-plane deformation, hence, in a fluid environment such as liquid, the influence of the viscous fluid damping on a BAW resonator can be reduced, and so does the acoustic energy loss. Combining these distinctive advantages, BAW resonators operating at lateral modes can achieve a high Q factor in air or liquid. Thus, BAW resonators have significant potential to be applied to chemical and biological sensing applications [29,30,31,32,33,34,35].

Combining the approach of coupled resonators with BAW resonators has recently led to achieving better performance and new functionalities. In particular, coupled BAW resonators based on lateral vibration modes are studied due to the advantages mentioned above.

The two main lateral vibration modes, the extensional/breathing mode [26] and the wine-glass (WG) mode [26], gain tremendous attention due to the potential to achieve relatively high Q factors. Their applications range from coupled resonators based filters [14,36] to oscillators [37,38] and sensors [39,40,41,42]. To characterize their sensitivity in the sensing applications, both the conventional frequency shift and, more recently, amplitude ratio shift based on mode localization are evaluated [43,44,45,46,47]. Devices using mode localization can show a considerably higher sensitivity compared to those using frequency shift as an output metric. Typically, the sensitivity is improved by two to four orders of magnitude, as demonstrated in the work on coupled resonators with different vibrational modes presented in [41,42,48,49,50,51]. Therefore, sensors based on coupled BAW resonators have been increasingly investigated as a promising approach for a range of sensing applications.

Here, we mainly focus on the theoretical and practical studies on coupled BAW MEMS resonators as sensors, oscillators, and filters. This paper consists of six sections: Section 2 introduces the fundamentals of coupled BAW MEMS resonators and their sensing mechanisms as sensors; Section 3 discusses the different transduction configurations applied to coupled BAW MEMS resonators; Section 4 reviews the previously reported work about coupled BAW MEMS resonators for different applications; Section 5 presents the conclusion and an outlook of coupled BAW MEMS resonators.

## 2. Fundamentals of BAW Resonators

### 2.1. BAW Resonators

#### 2.1.1. BAW Propagation

There are two basic types of acoustic waves: SAW and BAW [4]. When the acoustic wave is confined and travels at the interface between the substrate and the air, it is called a surface acoustic wave (SAW), like the ripples observed on the water surface hit by a stone. In contrast, a wave that propagates in the bulk of the material of the medium and exists in most of the volume of the media, is called a bulk acoustic wave (BAW), like a sound wave that travels through the air.

BAWs mainly refer to elastic waves that propagate in the solid medium and are mainly categorized into the following types: longitudinal waves [4], transverse (shear) waves [4], Rayleigh waves [4], and lamb waves [4,25]. The Rayleigh wave is a combination of longitudinal and shear waves. In Rayleigh waves, surface particles of the isotropic solid move in ellipses in the plane normal to the surface and parallel to the wave direction [4]. Table 1 shows a description of the other three BAW wave types.

BAW resonators are a type of MEMS device experiencing acoustic wave propagation through the bulk of the medium. Currently, there are two main types of BAW resonators applied in the industry: thin-film BAW resonators (FBARs) [52,53,54] and solidly mounted resonators (SMRs) [54,55,56]. These two types of resonators have the same working principles but different fabrication technologies.

#### 2.1.2. Bulk Vibration Modes

Based on the different acoustic wave propagation, BAW resonators vibrate with two types of motion: the in-plane motion and the out-of-plane motion. The in-plane vibrational modes mainly include the longitudinal mode and the anti-symmetric lamb mode, also called the flexural or bending mode [4], while the out-of-plane vibrational mode includes the transverse mode and the symmetric lamb mode, also known as the lateral bulk mode [26].

Note that the resonance frequency in BAW devices operating at the flexural (out-of-plane) mode is related and limited to the thickness of the device similar to the FBARs operating in the longitudinal mode [4]. Compared to flexural-mode devices, bulk-mode devices utilizing lateral in-plane vibration modes have a higher stiffness because of their higher bulk moduli [9]. Therefore, they vibrate at higher operational frequencies and are consequently less prone to internal mechanical losses, such as thermoelastic damping (TED), thereby achieving larger Q factors.

Over the past decades, two types of lateral bulk vibration modes have been mainly investigated for BAW resonators, with applications as oscillators, sensors, and filters: (i) the extensional (breathing) mode and the (ii) the WG mode of square-plate or disk structures [26,57,58,59,60,61,62,63,64]. Figure 1 and Figure 2 show a square plate and a disk BAW resonator, respectively, both exhibiting these two modes. Taking a square-plate BAW resonator as an example, the plate compresses and extends asymmetrically on the opposite sides in the WG mode, while it compresses and extends symmetrically on all four sides in the extensional mode [26].

It has been shown that the fundamental WG mode (WG1) can achieve a higher Q factor than the extensional mode [26]. A BAW resonator in air and even in liquid operating in higher-order WG modes (e.g., 2nd order WG mode [65] or the 3rd order WG mode [66]) usually has a higher Q factor than those working in the WG1 mode. Besides, for a disk resonator working in higher-order radial modes [67], the anchor loss is no longer the dominant source of energy dissipation. Other modes are also possible; Figure 3 shows some other bulk modes, such as the button-like (BL) [65], face-shear [68], and butterfly modes [69].

#### 2.1.3. BAW Resonators with Different Suspension Structures

The Q factor of a resonator is a dimensionless parameter to describe the loss of the vibration energy, defined by the ratio of the total energy stored with respect to the energy dissipated per cycle [26]. The energy losses can originate from different sources of energy dissipation. The clamping loss at anchors, also known simply as anchor loss, is a significant source of the energy loss mechanism, in particular when the anchors are stressed at the clamping points due to the displacement of the resonator during vibration. A fraction of the vibration energy stored is transferred to the substrate through the suspension stems and anchors [70,71].

The suspension structure also has a strong influence on the Q factor and also on the choice of the fabrication process. Two commonly used side-clamped suspensions, the simple straight-beam [26] and T-shaped suspensions [26,72] have already been widely employed in BAW resonators, as illustrated in Figure 4. The strain energy stored in the straight-beam suspension is related to the whole stem part. The strain energy stored in the T-shaped suspension is mostly associated with the cap because the stem part is decoupled from the anchor via the cap [26]. The ratio of the strain energy stored in the vibrating structure over the strain energy stored in the suspension part can be regarded as an indicator of anchor loss. The higher this ratio, the lower Q deduction induced by the anchor loss, especially when the anchor loss is the dominant energy dissipation mechanism in the system [26]. A method of calculating the strain distribution in the suspension part and resonator body using ANSYS FEA is proposed in [26,72], which could be helpful to design proper suspension structures.

Besides, the structure of the device can be clamped at different positions, such as the center (for surface micromachined structures) [73], the sides [68], and the corners [26,74] of the structure, as illustrated in Figure 5). How to choose the proper positions of the suspensions mainly depends on the minimum motion of working modes (to reduce the clamping loss as much as possible) and the fabrication possibility. Due to the manufacturing constraints, the sides and corners of the vibrating structure are commonly used locations for suspensions connected to the anchor, especially for designs with the backside substrate etched off for the releasing purpose. As for the devices with suspensions at the center, it is difficult to manage to fabricate devices with perfect suspension alignment. Perfect suspension alignment cannot guarantee high Q factors, for the suspension with finite dimensions will also restrict the motion outside the center part, which results in further energy losses [73]. Besides, Q factors are deduced with the mode number increasing since the device center is closer to the high-velocity points at higher modes. The misaligned suspension will couple the resonator vibration to the substrate, then the energy is lost to the substrate, thus the overall Q factor is degraded [73].

A study on the resonators with a different number of suspensions [75], indicates that a BAW resonator can vibrate in the expected modes even with a minimum of two suspensions. The anchor losses generally are lower with a low number of suspensions. However, the resonance frequency also decreases with a small number of suspensions, as fewer suspensions result in a decrease in the effective stiffness.

### 2.2. Multi Degrees-of-Freedom (Multi-DoF) Coupled BAW Resonators

#### 2.2.1. Mass-Spring-Damper System for a Single Degree of Freedom (1-DoF) BAW Resonator

A single MEMS resonator can be modeled as a mechanical mass-spring-damper system with 1-DoF. As shown in Figure 6. *M*, *K*, and *c* represent the mass, the supporting spring stiffness, and the damping coefficient of the resonator, respectively.

Assuming *F* = 0 and *c* = 0, the natural frequency of this system can be obtained, expressed as [48]:(1)$$f_{0}=\frac{1}{2\pi }\sqrt{\frac{K}{M}}$$

For a BAW resonator with a square plate structure using lateral vibration mode, the effective mass can be expressed as:(2)$$M_{eff}=\rho hL^{2}$$

The effective stiffness is given by:(3)$$K_{eff}{=\pi }^{2}Eh$$

Therefore, the resonant frequency of the lateral modes like SE and WG modes can be approximately obtained by:(4)$$f_{0}=\frac{1}{2L}\sqrt{\frac{E}{\rho }}$$
where *L* is the length of the square plate, *h* is the thickness, *ρ* denotes the density of the material of the plate, and *E* represents Young’s modulus.

#### 2.2.2. Coupling Mechanisms

For multi-DoF BAW resonators, mechanical coupling in form of a mechanical spring or electrostatic coupling in form of an electrostatic spring can be used as a coupling mechanism [48,76], and this is illustrated in Figure 7. Both approaches can be regarded as equivalent in theoretical calculations and achieve the same level of effective coupling spring stiffness. When the coupling stiffness is much smaller than the stiffness of the suspension of the resonators, a weakly coupled system is formed. As for the strongly coupled system, if the coupling beam is set to be an integer number of half-wavelengths and the coupled system is operated at high frequencies, a very strong coupling element is achieved between the two resonators [77].

However, the strong coupling does not apply to the VLF-MF flexural mode resonators, for at those high frequencies, the length of the coupling beam should reach hundreds of microns to be equal to the half-wavelengths, which is impractical for the real dimension design [77]. So far, there is less relevant work on the strongly coupled BAW resonators, and there is still no systematic theoretical model built. Therefore, in this review, the theory of the weakly coupled system will be mainly discussed. For the weakly coupled resonators, due to the fabrication tolerance, the structure likely becomes asymmetric, thus a pre-mode localization (discussed in Section 2.3) will occur. This caused a non-uniform energy distribution, but the structure symmetry still can be adjusted [77]. Moreover, a weakly coupled system will lead to a significant increase in sensitivity when coupled BAW resonators are used as sensors, and this will be mainly discussed in Section 2.3.

Two conventional designs of a mechanical coupling spring are the fixed-fixed guided beam and the folded coupling beam [76], as shown in Figure 7a and b respectively. The geometric parameters of the mechanical beams can be chosen to achieve different coupling strengths. The higher the ratio between length and width of a mechanical coupling spring, the weaker the coupling strength. Besides, the position of the coupling spring also has an obvious impact on the coupling strength, as shown in Figure 8.

Figure 8 shows FEM simulated WG modes of a 2-DoF coupled BAW resonator system with a mechanical coupling beam of the same dimensions but located at different positions. The dimensions and material selection are listed in Table 2. The devices have different resonant frequencies of the in-phase and out-of-phase WG modes [78]. The coupling strength has an important influence on the frequency difference between the in-phase and out-of-phase modes of the 2-DoF resonator. A small coupling strength results in a small frequency difference between in-phase and out-of-phase modes as summarized in Table 1. As shown in Table 3, if the coupling beam is closer to the top or bottom sides of the proof mass, i.e., the square plate shown in Figure 8, the former in-phase mode frequency increases while the latter out-of-mode frequency decreases, and this means the frequency difference between these two modes decreases, and so does the coupling strength. The relationship between the frequency difference and the coupling strength for the 2-DoF coupled resonators can be calculated by Equations (4) and (5) (see Section 2.2.3). Therefore, weak coupling can be achieved by adjusting the position of coupling beams without the need for changing the dimensions of the beams.

A detailed study on 2-DoF weakly coupled square-plate BAW resonators [41,42] indicated that with a short mechanical coupling beam that is designed away from the center position of the proof mass, weak coupling can be achieved. For the coupled BAW resonators, different pairs of in-phase and out-of-phase modes exhibit different coupling stiffness. Optimal in-phase and out-of-phase modes should have enough frequency difference to distinguish them from each other to avoid mode aliasing [50] (this will be discussed further in Section 2.2.5). Therefore, there is a trade-off between weak coupling strength and frequency spacing.

The electrostatic coupling (shown in Figure 7c) is achieved by applying a direct current (DC) voltage across a capacitive gap located between the two resonators [76]. The displacement-dependent component of the generated electrostatic force between the two resonators results in a spring-like behavior. The extension or compression direction of the electrostatic spring is aligned to the movement direction of the resonators, resulting in an elastic spring with negative effective spring stiffness. Note that the coupling strength can be easily tuned and controlled by changing the applied DC voltage. A very weak coupling can be achieved by applying a small DC voltage, which is much more convenient compared to mechanical coupling. However, the electrostatic coupling can be well applied to those coupled resonators with a low suspension spring, such as with the coupled double-ended tuning fork (DETF) resonators [49,50,51], where the mode aliasing effect can be avoided within a realistic range of DC voltages. Compared to DETF resonators [49,50,51], coupled BAW resonators using the lateral bulk mode (e.g., WG mode) have much higher effective stiffness, and thus a higher resonant frequency can be achieved. The electrostatic coupling strength (even the coupling voltage difference is under 300 V) is too small compared with the suspension spring stiffness of BAW resonators. That makes the electrostatically coupled BAW resonators prone to have the mode aliasing effect even with a high Q factor, as the generated frequency difference between in-phase and out-of-phase modes is too small to be told apart, within the working DC voltages.

Compared with the electrostatic coupling, the mechanical coupling has a higher stiffness (still smaller but not negligible compared with the suspension stiffness) and can be designed to avoid the mode aliasing and fulfill the requirement of the weak coupling at the same time. That is the reason why there is still no relevant research about the electrostatically coupled BAW resonators using the lateral vibration modes (the WG mode and extensional mode).

Figure 9 shows different vibration modes of a 2-DoF electrostatically coupled disk BAW resonator system. The dimensions and material selection are listed in Table 4. Each resonator has a short convex plate parallel to that of the other resonator to achieve electrostatic coupling, and a small potential difference of 100 V is applied to these two resonators. Two vibration modes are studied: Mode 1 with lower resonant frequencies and mode 2 (WG mode) with higher resonant frequencies. Table 5 summarizes their frequencies and shows that the frequency difference of WG mode 2 is about three orders of magnitude lower than that of mode 1. In conclusion, the mode with higher frequencies is more prone to mode aliasing than that operated at lower frequencies.

#### 2.2.3. Mass-Spring-Damper System for 2-DoF Weakly Coupled BAW Resonators

As shown in Figure 10, a 2-DoF weakly coupled BAW resonator system is modeled as a mass-spring-damper system [48], where *M*_1_ and *M*_2_ are the mass of two resonators, *K*_1_ and *K*_2_ are the suspension spring stiffness, *K*_c_ is the stiffness of the coupling spring, *c*_1_ and *c*_2_ are the damping coefficient. Similarly, the displacements of the two resonators are *X*_1_ and *X*_2_, and the external forces applied to the two resonators are *F*_1_ and *F*_2_.

Assuming *F*_1_ = *F*_2_ = 0 and *c* = 0, the resonant frequency of this system can be obtained, expressed as [48]:(5)$$f_{1}=\frac{1}{2\pi }\sqrt{\frac{K}{M}}$$
(6)$$f_{2}=\frac{1}{2\pi }\sqrt{\frac{K+2K_{c}}{M}}$$

#### 2.2.4. Mass-Spring-Damper System for 3-DoF Weakly Coupled BAW Resonators

Figure 11 shows a 3-DoF weakly coupled BAW resonator system [48], *M*_3_ is the mass of resonator 3, *K*_3_ is the supporting spring stiffness, *Kc*_1_ and *Kc*_2_ are the stiffness of the coupling spring, and *c*_3_ is the damping. Similarly, the displacement of resonator 3 is *X*_3_, and the force applied to the three resonators is *F*_1_, *F*_2_, and *F*_3_, respectively.

Assuming *F*_1_ = *F*_2_ = *F*_3_ = 0 and *c* = 0, the resonant frequency of this system can be obtained, expressed as [79]:(7)$$f_{1}=\frac{1}{2\pi }\sqrt{\frac{K}{M}}$$
(8)$$f_{2}=\frac{1}{2\pi }\sqrt{\frac{K+K_{c}}{M}}$$
(9)$$f_{3}=\frac{1}{2\pi }\sqrt{\frac{K+3K_{c}}{M}}$$

#### 2.2.5. Mode Aliasing

Unlike 1-DoF resonators, coupled resonators can exhibit multiple modes: 2-DoF coupled resonators normally exhibit two modes: in-phase and out-of-phase modes, and 3-DoF coupled resonators can exhibit three different in-phase and out-of-phase modes [48,49,50,51]. Different modes have different resonant frequencies as well as different working performances, such as the sensitivity performance when resonators serve as sensors.

However, due to the finite bandwidth of each mode, if the frequency difference between the adjacent two modes is too small, the two modes will interfere and merge, which results in only one observable resonant peak, and is called mode aliasing [50]. Therefore, once mode aliasing happens, the multiple modes of coupled resonators cannot be told apart and let along to use the characteristic of two modes for the sensing purpose. To reduce and avoid mode aliasing, the structure of resonators (especially the coupling strength) should be designed to achieve a large enough frequency difference between two adjacent modes, satisfying [50]:(10)$$\left(f_{2}-f_{1}\right)\geq 2f_{3\mathrm{dB}}$$
where *f*_3dB_ is the 3 dB bandwidth of each mode, *f*_1_ and *f*_2_ are the resonant frequency of two adjacent modes.

Additionally, different pairs of in-phase and out-of-phase vibration modes have different frequency differences. The WG mode with a relatively high frequency described above is more prone to the effect of mode aliasing than the mode with a lower frequency [48,49,50,51], as mentioned in Section 2.2.2, and the frequency differences of the mechanically coupled resonators operated in two modes (the same as the modes shown in Figure 9) are listed in Table 6. The data show that the frequency difference of mode 1 is much larger (more than 100 times larger) than that of WG mode, and this should be taken into consideration when you choose the proper working mode for your design of coupled BAW resonators, but currently, there is still no relevant study about this.

### 2.3. Weakly Coupled BAW Resonators

#### 2.3.1. Mode Localization

Weakly coupled BAW resonators normally indicate a situation where the coupling strength is far less than the supporting spring stiffness [48,80]. For the weakly coupled BAW resonators where two resonators are the same and there is no perturbation present, then the system is in balance, and the two resonators normally exhibit the same displacement. However, when a small external perturbation is introduced, the system will be imbalanced, and the vibrational energy will be mainly confined to only one resonator, the so-called mode localization phenomenon [43]. It has been proven that the sensitivity of weakly coupled resonators using the amplitude ratio |*X*_1_/*X*_2_| of two resonators as an output metric is several orders of magnitude higher than that using the conventional frequency shift as an output metric, which provides a new sensing mechanism with high sensitivity. As a result of the increase of the parametric sensitivity, the resolution limit of the mode-localized sensors can also be improved by orders of magnitude especially when more than 2-DoF resonators are coupled together, which has first been theoretically validated in [81].

Figure 12 shows two disk resonators that are weakly coupled together by a mechanical beam. The dimensions and material selection are listed in Table 7. For such a system, there are two types of vibration modes: the out-of-phase mode and the in-phase mode. Once a small mass perturbation is introduced into one of the resonators, the vibration amplitude [48] of the two resonators changes at the same time. The ratio |*X*_1_/*X*_2_| of the resonator amplitude at the same resonant frequency is then utilized for sensing applications. In Figure 13, the relationship between the relative mass change on resonator 2 and the amplitude ratio |*X*_1_/*X*_2_| is presented. When there is no mass perturbation in this system, the amplitude ratio is equal to 1, indicating both resonators have the same deformation. in the presence of a mass perturbation, the amplitude ratio is no longer equal to 1. Besides, the in-phase and out-of-phase modes have different mode shape changes with the same mass perturbation, which could be used to detect the position of the perturbation (for details see Section 2.3.4).

#### 2.3.2. Sensitivity Characterization

When there is a change in mass or stiffness in the resonator system, i.e., ∆*M* or ∆*K*, the resonant frequency will change. Conventionally, the resonant frequency shift with respect to the stiffness and mass perturbation of the system is utilized to indicate the sensitivity of a resonator.

For the 1-DoF BAW resonators, when ∆*M* is far less than *M*, i.e., ∆*M* ≪ *M*, which is the same as ∆*K* ≪ *K*, the normalized sensitivity [82] based on the frequency shift with respect to a small Δ*M* or Δ*K* (*M* = *M* + Δ*M* or *K* = *K* + Δ*K*) is obtained as [48]:(11)$$\left|S_{M}^{1DoF,\;f}\right|=\left|S_{K}^{1DoF,\;f}\right|=\left|\frac{\partial \omega_{0}}{\partial M\left(K\right)}/\frac{\omega_{0}}{M\left(K\right)}\right|\approx \frac{1}{2}$$

Above all, when a 1-DoF system is exposed to a small perturbation, i.e., the stiffness or mass perturbation, there will be a 50% variation in the output frequency.

For the 2-DoF BAW resonator, the frequency shift with respect to a small Δ*M* or Δ*K* on resonator 2 (*M*_2_ = *M* + Δ*M* or *K*_2_ = *K* + Δ*K*) is used to characterize the normalized sensitivity which is obtained as:(12)$$\left|S_{M}^{2DoF,\;f}\right|=\left|S_{K}^{2DoF,\;f}\right|\approx \frac{1}{2}$$

For the 2-DoF BAW resonator based on the mode localization effect, the normalized sensitivity with respect to a small Δ*M* or Δ*K* on resonator 2 (*M*_2_ = *M* + Δ*M* or *K*_2_ = *K* + Δ*K*) represented by the amplitude change (*X*) is obtained as [48]:(13)$$\left|S_{M}^{2DoF,\;X}\right|=\left|S_{K}^{2DoF,\;X}\right|=\left|\frac{\partial \left|\Delta X/X\right|}{\partial M\left(K\right)}/\frac{\left|\Delta X/X\right|}{M\left(K\right)}\right|\approx \frac{K}{4K_{c}}$$

Therefore, in terms of the amplitude change, the normalized sensitivity of the 2-DoF resonator is *K*/(2*K*_c_) times larger than that of the 1-DoF resonator. However, when the amplitude ratio (*AR*) of the two resonators is adopted as an index to characterize the normalized sensitivity, the normalized sensitivity represented by the *AR* is obtained as:(14)$$\left|S_{M}^{2DoF,\;AR}\right|=\left|S_{K}^{2DoF,\;AR}\right|=\left|\frac{\partial AR}{\partial \left(\Delta M\left(K\right)/M\left(K\right)\right)}\right|\approx \frac{K}{2K_{c}}$$
where *AR* = |*X*_1_/*X*_2_|.

Compared to the normalized sensitivity obtained by the amplitude change, the normalized sensitivity is improved by 2 times with the usage of *AR* as the output metric. Thus, *AR* is widely taken as the output metric when the weakly coupled resonators based on mode localization are used as sensors, i.e., the mode-localized sensors.

For the 3-DoF mode localized BAW resonator as sensors, the normalized sensitivity with respect to a small Δ*M* or Δ*K* on resonator 3 (*M*_3_ = *M* + Δ*M* or *K*_3_ = *K* + Δ*K*) is obtained as [48,49,50,51]:(15)$$\left|S_{M}^{3DoF,\;AR}\right|\approx \frac{K}{K_{c}}+1$$
(16)$$\left|S_{M}^{3DoF,\;AR}\right|\approx \frac{K}{K_{c}}$$
where *AR* = |*X*_1_/*X*_3_|.

The normalized sensitivity of the mode localized 3-DoF sensor is about two times higher than that of the 2-DoF resonator mode localized sensor, but if the stiffness *K*_2_ is designed to be two times or larger than *K* (*K*c < *K*/10 < *K*_2_/20, *K*_1_ = *K*_3_ = *K*) [48,49,50,51], the normalized sensitivity can be expressed as:(17)$$\left|S_{M}^{3DoF,\;AR}\right|\approx \frac{\left(K+K_{c}\right)\left(K_{2}-K+K_{c}\right)}{K_{c}^{2}}$$
(18)$$\left|S_{K}^{3DoF,\;AR}\right|\approx \frac{K\left(K_{2}-K+K_{c}\right)}{K_{c}^{2}}$$

The 3-DoF mode localized sensor has a sensitivity with the *AR* output which is at least 48 times higher than that of the 2-DoF resonator based on mode localization. Thus, there is plenty of room for adjusting the parameters of the coupled resonators to improve their sensitivity.

#### 2.3.3. Common Mode Rejection

As for coupled BAW resonators with two identical resonators, any change in the ambient condition like a temperature fluctuation or a pressure fluctuation, causes a change in both resonators simultaneously, which is called the common-mode change [83,84]. The theoretical model of the temperature characteristics of the amplitude ratio and frequency of the mode-localized sensors under the influence of Young’s modulus has been established to reveal the common more-rejection property of the mode-localized sensors [85].

For example, in a 2-DoF weakly coupled resonator system with two identical resonators, the temperature (*T*) introduced stiffness change is Δ*K_T_*, which is applied to both resonators as a common-mode signal. Now let’s consider a situation in which a small external stiffness perturbation (Δ*K*) is applied to only one of the two resonators (e.g., resonator 2 here) in the presence of Δ*K_T_*:(19)$$K_{1}=K+\Delta K_{T}$$
(20)$$K_{2}=\left(K+\Delta K_{T}\right)+\Delta K$$

Therefore, the sensitivity based on the frequency shift or the AR of the coupled resonators with two identical resonators is only related to the added stiffness perturbation (Δ*K*) and the influence of Δ*K*_T_ on the sensitivity is canceled out as a common-mode signal. However, in reality, the two resonators are not fully identical due to fabrication errors, and the influence of Δ*K*_T_ on the sensitivity (the frequency shift or the *AR*) can be reduced dramatically but not completely [83], as shown in Figure 14.

Note that in the system of the 1-DoF resonator, there is only one resonator. Thus, when Δ*K_T_* and Δ*K* are both introduced to the resonator at the same time, the system stiffness *K*_s_ is:(21)$$K_{S}=K+\Delta K+\Delta K_{T}$$

The frequency shift is:(22)$$\Delta f\left|{}_{\Delta K+\Delta K_{T}}\right.\approx -\pi \frac{\Delta K+\Delta K_{T}}{K}\times f_{0}$$

In this case, it is hardly clear whether the frequency shift is caused by the Δ*K_T_* or caused by the Δ*K*, making the sensors based on the 1-DoF resonator vulnerable to temperature fluctuation if no extra temperature compensation method is used.

#### 2.3.4. Detection of the Position of Perturbations

For the coupled resonator, as mentioned above, the 2-DoF resonators normally have two different modes: in-phase and out-of-phase modes, and the 3-DoF coupled resonators can exhibit three different modes. Taking a 2-DoF weakly coupled BAW resonator system as an example, the relationship between the AR and mass perturbations on the resonator 2 is different when the resonator works in the in-phase or out-of-phase mode, as shown in Figure 12 and Figure 13. That is the case for the 3-DoF weakly coupled resonators as well. To conclude, to verify the position of the perturbation, the weakly coupled resonators can be driven selectively at their in-phase or out-of-phase mode, and then the corresponding ARs can be measured to locate the position of the perturbation accurately. Alternatively, researchers can employ the anti-resonances in the amplitude-frequency responses to identify which resonator is applied with a mass perturbation [86].

## 3. Coupled BAW Resonators with Different Transduction Methods

Actuators and sensing elements are vital components of MEMS resonators. For coupled BAW resonators, capacitive actuation and sensing elements, piezoresistive sensing elements, and piezoelectric actuation and sensing elements are commonly adopted. Table 8 provides an overview of these several transduction methods including a summary of their theoretical calculation expressions of actuation force and sensed motional current, and a comparison among them to show their pros and cons.

### 3.1. Capacitive Actuation and Sensing

Parallel plates like comb fingers are normally utilized to implement the capacitive actuation [1,87,88,89,90,91]. In the capacitive actuation, two parallel plates are loaded with an actuation voltage to form an electric field and generate an electrostatic force.

In terms of capacitive sensing [89,90,91], the displacement of the moving plate causes a change in the capacitance, leading to the charge flow of the capacitor, resulting in a motional current.

Capacitive transduction is a popular method for actuation and sensing of a MEMS device due to its low-cost and flexible design, simplicity of fabrication and implementation, and the capability to integrate with interface electronics systems [76,92,93]. However, it also has some drawbacks such as nonlinearity [91], pull-in instability [94], and an AC force that could appear at the double frequency, the feedthrough signal.

**Table 8 sensors-22-03857-t008:** Description of several classic transduction mechanisms applied to coupled BAW MEMS resonators.

Transduction Methods	Capacitive Actuation and Sensing	Piezoresistive Sensing	Piezoelectric Actuation and Sensing
Actuation force (*F*)	$F\approx \left(\frac{1}{2}V_{d}^{2}+2\left|V_{\mathrm{ac}}\right|V_{d}\right)\frac{dC}{dx}$ [76]	Not applicable yet	$F=\frac{e_{33}A_{3}V_{ac}}{h}-\;\mathrm{Longitudinal}$ $F=\frac{e_{31}A_{1}V_{ac}}{h}$- Transverse [92]
Sensed motional current (*i*_mot_)	$i_{\mathrm{mot}}=V_{d}\frac{\partial C}{\partial x}\frac{\partial x}{\partial t}$ [76]	$i_{\mathrm{mot}}\approx \frac{V_{d}}{R}\left(\frac{\Delta R_{r}}{R_{r}}\right)$ [95]	$\left|i_{\mathrm{mot}}\right|=\left|\omega \cdot e_{33}A_{3}S_{3}\right|\;\mathrm{Longitudinal}$ $\left|i_{\mathrm{mot}}\right|=\left|\omega \cdot e_{31}A_{3}S_{1}\right|$ Transverse [92]
Cons	Parasitic capacitance; Nonlinearity; complex circuit design; thin film damping; limitation in liquids [96]	Temperature dependency; high power consumption with using high DC voltages; noisy	Processing difficulty in piezoelectric materials; not possible static measurement
Pros	Low power consumption; good noise performance; easy to fabricate	Simple setup; inherent shielding; applicable to liquids [97]	Eliminating frequency drifts caused by DC voltage variations due to no DC voltages needed here [13]; Applicable to liquids [98,99,100,101]

#### 3.1.1. One-Port Configuration

In the one-port configuration [95] as shown in Figure 15, each resonator has four capacitive electrodes all serving as actuation electrodes, and the motional current is picked up from the anchors of the resonators. For the one-port configuration, the motional current can be increased by increasing the electrode overlap area and decreasing the gap between the electrode and the resonator body. However, the feedthrough current [102,103,104] can also be increased by shrinking the gap.

#### 3.1.2. Two-Port Configuration

As illustrated in Figure 16, in a two-port configuration [95,104], each resonator has one group of electrodes for actuation and the other group of electrodes for sensing. Compared to the one-port configuration, under the same actuation voltage, the actuation force decreases but the feedthrough signal decreases in the two-port configuration as the sensing and driving electrodes are separated and cross-talk capacitance is reduced.

### 3.2. Capacitive Actuation and Piezoresisitive Sensing

The piezoresistive effect can occur in all materials, which refers to the change of resistance resulting from mechanical stress. This effect has been applied to many commercial devices such as pressure sensors and accelerometers [1]. For the pressure sensor, integrated piezo resistors are used to measure the deformation of the pressure-sensing membrane. For the piezoresistive accelerometer, the resistance of piezo resistors embedded in the supporting springs changes with the stress variation due to the input acceleration, which is utilized to measure the input acceleration. The resistance measurement is also easy to be implemented and the piezo resistors are inherently shielded structures, which make it feasible and popular for the application of microsensors. As shown in Figure 17, a DC voltage (*V_d_*) is applied to the resonator at the anchors to detect the change of resistance, and then the motional current can be obtained [39,40].

In piezoresistive sensing configurations [105,106,107,108,109], all surrounding electrodes can be used for capacitive actuation because the motional current is picked up from the body of the resonator. Besides this, the motion signal can be increased by tuning *V_d_* rather than shrinking the transduction gap or increasing the applied actuation voltage, but it is limited to the piezoresistive coefficient of the single crystal silicon and the dissipation of the electric power of the resonator.

For BAW resonators vibrating at the extensional mode, the motional current is larger with the piezoresistive sensing configuration than with the capacitive sensing configuration [95]. Figure 18 shows the frequency responses for a square-plate BAW resonator operated in the extensional mode with different sensing methods but the same actuation. In contrast to the device with capacitive sensing, the device with piezoresistive sensing improves the output signal from 0.02 dB to 0.5 dB, and the signal distortion caused by the feedthrough signal is eliminated [39,40].

In another study, Iqbal et al. investigated the 2-DoF and 3-DoF square-plate and 2-DoF disk BAW resonators in the extensional mode [37,38]. In this work, the total output current was enhanced by summing the currents of all resonators and the piezoresistive and capacitive sensing methods were both adopted to be compared with each other. In [38], the insertion loss and the Q factor of a 2-DoF resonator using the piezoresistive sensing were compared with that of a 1-DoF resonator using capacitive sensing. There was a 10 dB reduction in the insertion loss and an increase of 3400 in the Q factor for the device using piezoresistive sensing proposed in [38]. Therefore, the results show that with the piezoresistive sensing configuration (shown in Figure 19 and Figure 20), the overall transduction efficiency was improved, thereby obtaining a higher Q factor.

### 3.3. Piezoelectric Actuation and Piezoelectric Sensing

The piezoelectric effect refers to the ability of certain materials which can generate the electric charge in response to the applied mechanical stress and vice versa (mechanical stress can result from an applied electrical field) [1]. There are two common piezoelectric transduction configurations [1,92]: the longitudinal configuration (shown in Figure 21a), where the force (*F*) is in the direction of the electric field (*ε*_3_), and the transverse actuation configuration (see Figure 21b), where the force (*F*) is perpendicular to the applied electric field (*ε*_3_). The electrodes (marked in grey in Figure 21) on the top and bottom surface of the piezoelectric material form a capacitor and the current through the capacitor will have a piezoelectric component in addition to the regular capacitance current.

The commonly used piezoelectric materials are PZT, ZnO, and AlN. Among these three materials, PZT has the highest piezoelectric coefficient but the lowest acoustic velocity which will result in more inner mechanical energy loss, thereby deducting the Q factor [96]. ZnO is chemically unstable, and this will lead to non-consistent measurement results. AlN has a lower piezoelectric coefficient but the highest acoustic energy, so there will be less inner mechanical loss [96]. The AlN thin film is insulated. Moreover, the compatibility with CMOS of AlN also makes it potentially applicable to chip-level integration [96]. The devices presented below are all fabricated using AlN as the piezoelectric material.

As an example, Chellasivalingam et al. designed two identical square-plate resonators weakly coupled by a short beam, and resonators were operated at the WG mode [41,42], as shown in Figure 22. The reverse piezoelectric effect was used for the actuation of the resonator while the piezoelectric effect was used for the pick-up of the motion. The piezoelectrically induced strain was converted back to an electrical output voltage. The mode localization was adopted in the weakly coupled BAW resonators to characterize its mass sensitivity. The normalized sensitivity based on the AR was 148.22, and the Q factor of 1773.8 was achieved in a vacuum. The fabrication of these two devices followed the PiezoMUMPs process [110]. The silicon device layer was directly utilized as the ground layer, then a sputtered AlN film on the Si layer served as the piezoelectric layer. Finally, a thin film of Al was deposited on the AlN layer as the top electrode, and thus the thin-film piezoelectric-on-silicon resonator (TPOS) was developed [111,112,113].

As piezoelectric resonant devices like FBARs exhibit high electromechanical coupling efficiencies without DC voltages, there will be a low signal transmission loss, thereby achieving a relatively small motional resistance, which is beneficial to the design of the setup for the measurement. Additionally, the DC voltage is not required for the operation of piezoelectric devices, which makes it more suitable for resonators operated in liquids than resonators with capacitive transduction. This is due to the DC voltage used for capacitive transduction in liquid being limited to several volts to avoid the electrolysis phenomenon in liquid. The electrolysis phenomenon will lead to a low transduction factor for capacitive sensing.

Therefore, combined with the advantages of the piezoelectric transduction and in-plane vibration modes, weakly coupled BAW resonators exhibit a high potential to be applied as sensors for chemical and biological applications. So far, the coupled BAW resonators based on mode localization have not been explored extensively.

### 3.4. Capacitive-Piezo Transduction

For the piezoelectric resonators shown in Figure 23a, both the top and bottom electrodes are directly in contact with the piezoelectric layer, and the interfacial strain loss normally occurs in the interface between the electrode and piezoelectric layers [1,28,42,92]. For the capacitive resonators indicated in Figure 23b, the transduction gaps are designed to separate the resonator body and electrodes, so there will be no interfacial strain loss [1,87,88,89,90,91]. Combining the main features of piezoelectric and capacitive resonators, a capacitive-piezo transduction configuration [109] where a resonator made of pure piezoelectric material (AlN) is separated from the electrodes by a small transduction gap is developed, as shown in Figure 23c. The piezoelectric component is used for achieving high coupling, and the non-contacting capacitive transduction component is used for achieving a high Q factor [109].

Figure 24 shows a 1-DoF and 2-DoF capacitive-piezo BAW disk resonant device with different working modes and anchors. In these two devices, a small transduction gap is designed to reduce the strain loss from the resonator body to the electrode and to eliminate the energy loss from the metal layer-to-piezoelectric layer interface. This gap should be small enough to make full use of the strong electromechanical coupling efficiency of the piezoelectric transduction, so a nano-scale gap is adopted here. For the 1-DoF resonator device developed by Robert et al. [115], a new self-switching resonator was achieved with a high Q of 9000 at 300 MHz enabling the device to act as a switchable AlN filter for RF front ends. Another study based on the 2-DoF resonator device [114], showed a higher Q factor of 12,748 which is more than 2.2 times higher than that of the conventional piezoelectric resonators with contacting electrodes.

## 4. Applications of Coupled BAW Resonators

### 4.1. Sensors Based on Weakly Coupled BAW Resonators

In the sensors based on the weakly coupled resonators, that are using the amplitude change or the AR change to measure the physical quantity change in the environment, the sensitivity will be largely enhanced compared to a sensor using resonant frequency changes as an output metric as illustrated in Section 2.3. The group of Professor Ashwin A. Seshia of Microsystems Technology in the Department of Engineering at Cambridge University focuses on the relevant study, and the related work is represented below.

Lin et al. developed a mass sensor based on the coupled BAW resonators [39,40] where the two square-plate BAW resonators were coupled together by a long and thin beam and operated in the extensional mode shown in Figure 25. In this device, one resonator was used for transduction while the other one was used for mass perturbation. A Cr film [39] was deposited on the surface of the resonator as a small mass perturbation on the system, and then a frequency shift (about 34 Hz/ng in air and 32 Hz/ng in vacuum for the in-phase mode; about 36.7 Hz/ng in air and 37.3 Hz/ng in vacuum for the out-of-phase mode) was observed due to the thin film deposition. Then to explore the feasibility of the coupled BAW resonators for biochemical sensing application, the streptavidin-coated polystyrene microbeads (SCPM) [40] and High Five insect cells [40] were subsequently applied to the functionalized surface of one resonator as a mass perturbation, shown in Figure 26.

Figure 27 shows the relationship between the frequency shift and the number of analytes attached to the resonator. Figure 27a shows a good match between the theory and the measurement result based on the analytes of SPCMs. In Figure 27b the frequency shift caused by the attached insect cells, −12 Hz/cell (−3.89 ppm/cell), is illustrated [40], indicating the potential of this device to serve as a biochemical sensor.

However, compared to the detection limit of the BAW resonators operated in liquid (i.e., SMR (0.01 ng/cm^2^) [116] and C-FBARs (1.78 ng/cm^2^) [117]), the detection limit of this device operating in the air (1.46 ng/cm^2^ or 36 ng/cm^2^) is not superior. To improve the sensitivity, further miniaturization of the proof mass or a secondary label of mass may be needed.

The conventional frequency shift is used to characterize the sensitivity in the work shown above, but for now, there have already been two studies from one group on the weakly coupled BAW resonators utilizing mode localization as the sensing mechanism. Chellasivalingam et al. [41,42] coupled two identical square-plate resonators by a short beam which are both operated in the WG mode (see Figure 24). The piezoelectric actuation and sensing mechanisms were adopted here.

The device proposed in [41] was used as a mass sensor to detect the Polystyrene Latex (PSL) particles with a diameter of about 296 nm. The results showed that the normalized sensitivity based on the AR shift (422) was much larger than that based on the conventional frequency shift (−0.17) in a vacuum as shown in Figure 28a. For the device proposed in [42], it was used for aerosol detection, and the soot particles of 100 nm in diameter were applied to the surface of one resonator as a mass perturbation. The normalized sensitivity obtained in a vacuum (0.0735 for frequency shift; 148.22 for amplitude ratio shift) is shown in Figure 28b. The minimum mass it can detect based on the AR and frequency shift is 367.8 pg and 2.16 pg, respectively.

Unlike the device described in [39,40], the sensitivity of the BAW resonators based on mode localization was improved considerably. However, as for the device proposed in the work [40], the minimum detection limit based on mode localization was much larger than that based on frequency shift which should be further researched.

To summarize, both frequency shift and mode localization have been proposed for characterizing the sensitivity of weakly coupled BAW resonators, and Table 9 shows a performance comparison of these reported works. The coupled BAW resonators based on mode localization showed great potential for enhanced sensitivity proved in theory and practical work shown above and worth further investigation.

### 4.2. Oscillators Based on Coupled BAW Resonators

The group of Professor Joshua E.-Y. Lee of the Department of Electronic Engineering at the City University of Hong Kong is carrying out the relevant study, and so is the group of Professor Clark T.-C. Nguyen in the Department of Electrical Engineering and Computer Sciences at the University of California at Berkeley. The related work is indicated below.

Iqbal et al. focused on the weakly coupled square-plate BAW resonators as oscillators, including 2-DoF and 3-DoF resonators, and resonators were diagonally coupled by a mechanical beam to make the piezoresistive sensing work the best [37], as shown in Figure 19. The 1-DoF, 2-DoF and 3-DoF resonators were demonstrated to work at the same resonant frequencies with the in-phase extensional mode, and the synchronized oscillation was realized via a coupling beam with a half-wavelength. Besides, with the introduction of the coupling mechanism, a higher Q up to 120,000 (3-DoF) was achieved compared to the Q of the single resonator (60,000), as illustrated in Figure 29a.

Iqbal et al. then studied a 2-DoF disk BAW resonator device as oscillators [38], and the synchronization was also achieved by a coupling beam at half of the wavelength (see Figure 20). The Q factor achieved here was about 184,000, two times higher Q factor than the Q of 1-DoF disk resonators, as illustrated in Figure 29b.

These two studies with similar results demonstrated the possibility of using weakly coupled BAW resonators with the extensional mode to work as an oscillator with an enhanced Q factor compared to single resonators. Based on this work, the weakly coupled BAW resonators operated at other lateral bulk vibration modes or using different transduction methods can be further researched to achieve a novel filter with a higher Q factor.

Besides, the device developed by Demirci et al. [73] has the potential to be an oscillator. As shown in Figure 30, there are two devices: a 3-DoF strongly coupled BAW resonator system with anchors at the center and a 7-DoF strongly coupled BAW resonator system with anchors connecting to the resonator body by a suspension beam, and they are operated in the out-of-plane transverse mode at 70 MHz.

By mechanically coupling resonators together, the total output current is boosted by combining the output current from each resonator in the coupled system, as shown in Figure 30c. The work in [73] indeed achieved a Q factor of more than 9000 and a reduction in the motional resistance (480 Ω, more than 5.9 times smaller than that of a 1-DoF resonator device) by mechanically coupling several resonators together, instead of shrinking the transduction gap between the electrode and the resonator body or increasing DC bias. In this way, the power handling capability was enhanced, the linearity of the system was maintained, and the motional resistance versus the dynamic range tradeoff often occurs when scaling was broken.

Lin et al. designed several multi-DoF strongly coupled disk filters operating at the first in-phase WG mode [118], and taking a 3-DoF system as an example, Figure 31a shows the disk resonators were strongly coupled by a mechanical beam with half-wavelength. Figure 31b indicates the operating frequencies were all around 61 MHz and the Q factors for 3-DoF, 5-DoF, and 9-DoF resonators were all about 120,000 (pretty high) although the single resonator could achieve a higher Q factor of 161,000. When the oscillator was divided down to 10 MHz, the phase noise of −138 dBc/Hz at 1 kHz offset and −151 dBc/Hz at far from carrier offset were achieved here, and there was a 13 dB and 4 dB improvement over the previously reported work on the single resonator-based oscillators [118], as shown in Figure 31c. That well verified the utility of coupled resonator-based filters which deserved to be further explored.

### 4.3. Filters Based on Coupled BAW Resonators

There is one study on the coupled BAW resonators based filters and the group of Professor Clark T.-C. Nguyen in the Department of Electrical Engineering and Computer Sciences at the University of California at Berkeley is carrying out the relevant research, and the related work is illustrated below.

Figure 32 shows a 2-DoF coupled BAW resonator system with anchors at the center of the resonator body, and the radial contour mode is chosen as the operation mode, consisting of out-of-phase and in-phase modes. Ozgurluk et al. studied the 2-DoF BAW resonator system as a filter, and to increase the output motional current, more resonators were then coupled and operated at the in-phase radial contour mode [80], shown in Figure 32. There were 96 disks strongly coupled by 110 mechanical beams, and the coupling beam is designed to equal half-wavelength [77]. Due to the possibility of structural asymmetry in weakly coupled structures, strong coupling is adopted here to ensure that all resonators vibrate uniformly at one mode frequency, hence the output currents of each resonator in the coupled system can sum constructively [80]. Besides, strong coupling can also eliminate non-uniform reductions in the motional resistance which normally occurs in the weakly coupled system [77]. Finally, a 0.1% bandwidth 223.4 MHz MEMS filter was achieved with a 2.7 dB insertion loss of the passband and a 50 dB stopband rejection.

Compared with the single disk resonator, the coupled disk resonators achieve a lower motional resistance. Figure 33 shows a comparison of frequency spectra between a 30-DoF disk system and a single disk system. The Q factor obtained for the 30-DoF disk resonator here was 8815, and the motional resistance (1180 Ω) was about 9 times lower than that of a filter based on a single disk resonator. That verified the feasibility of using coupled-resonator design.

A small transduction gap of 39 nm (even reach to below 10 nm) was adopted in this work to reduce the motional resistance, thereby improving the electromechanical efficiency, which was a landmark for the capacitively transduced MEMS resonators. Besides, the weakly coupled MEMS BAW resonator devices in [37,38,73] illustrated in Section 4.3 can also be further studied as filters with the needs of the actual RF channel-selecting receiver front-ends [80].

## 5. Conclusions and Outlook

The distinctive advantages of the BAW MEMS resonator over its counterparts include its small size, high stiffness, relatively high operating frequencies, large ratio of surface to volume, high Q factors, and high sensitivity as sensors, making it more suitable for some applications such as sensing, timing reference, and filtering applications. Moreover, BAW resonators with different operating modes, coupling methods, and transduction configurations can exhibit distinct work performances, and their respective advantages can be utilized to achieve different application purposes.

This review mainly presents the relevant studies on coupled BAW resonators used as oscillators, sensors as well as filters. Especially, the coupled BAW resonators with lateral vibrational modes using piezoelectric transduction are a good choice for chemical and biological applications (detecting some analytes such as proteins, DNA, and cells in liquid), since they are less prone to fluid damping. Furthermore, the piezoelectric configuration should be a good choice for the coupled BAW resonators, for it does not require DC voltages for operation. DC voltages are needed in the capacitive configuration to enhance the transduction factor but should be limited to only several volts in liquid to avoid electrolysis, limiting the electromechanical transduction efficiency.

However, there are still several issues that should be considered while designing devices based on coupled BAW resonators: (1) if serving as filters and oscillators, it is necessary to achieve low motional resistance, which requires achieving high Q factors and effective electromechanical transduction [1]; (2) if serving as sensors, there should be a compromise between the sensitivity and mode aliasing effect. The coupling strength should be designed reasonably, and the optimized structure to achieve a high Q factor is also significant. Besides, fabrication constraints should also be considered for designing your structure. Therefore, there is still lots of work to be done on the coupled BAW resonators. As different types of BAW resonators have their respective advantages, they can be further explored to improve their working performances for a wide range of applications in the following aspects: (1) coupled BAW resonators with other vibrational modes; (2) coupled BAW resonators using different coupling elements; (3) coupled BAW resonators utilizing different transduction methods to reduce the feedthrough current; (4) different piezoelectric materials chosen for the coupled BAW resonators using piezoelectric transduction with the TPOS structure; (5) structural optimization of the coupled BAW resonators to eliminate the effect of mode aliasing.

## Figures and Tables

**Figure 1 sensors-22-03857-f001:**
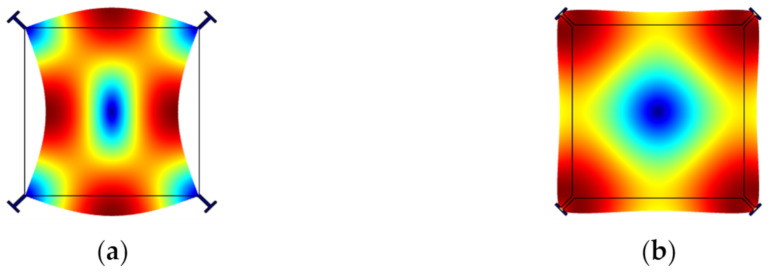
Finite element model (FEM) simulation of a BAW resonator with square plate structure in COMSOL. (**a**) WG mode; (**b**) Extensional mode.

**Figure 2 sensors-22-03857-f002:**
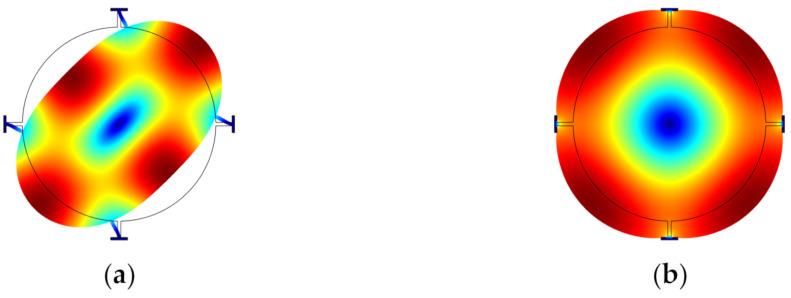
FEM simulation of a BAW resonator with disk structure in COMSOL. (**a**) WG mode; (**b**) Extensional mode.

**Figure 3 sensors-22-03857-f003:**
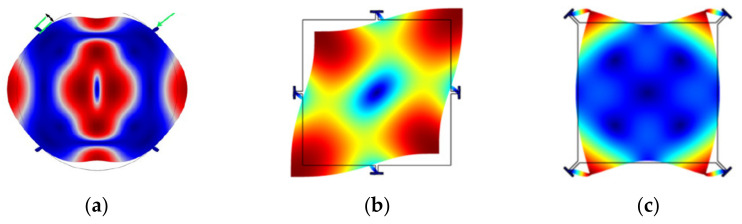
Other FEM simulated bulk modes in COMSOL. (**a**) BL mode (Adapted with permission from Ref. [63]); (**b**) Face-shear mode; (**c**) Butterfly mode.

**Figure 4 sensors-22-03857-f004:**
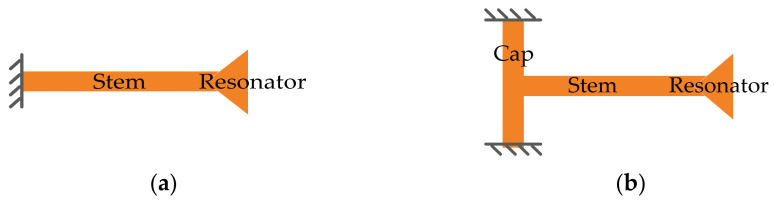
Schematic diagrams of suspensions. (**a**) Straight-beam suspension; (**b**) T-shaped suspension. Adapted with permission from Ref. [26].

**Figure 5 sensors-22-03857-f005:**
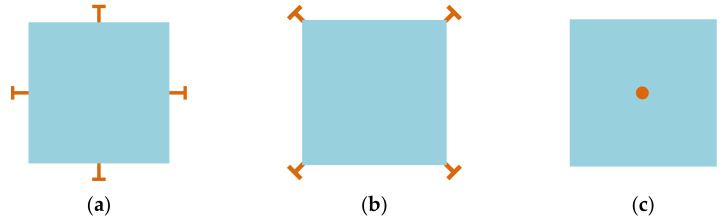
Suspensions at different positions: (**a**) On sides; (**b**) At corners; (**c**) At the center.

**Figure 6 sensors-22-03857-f006:**
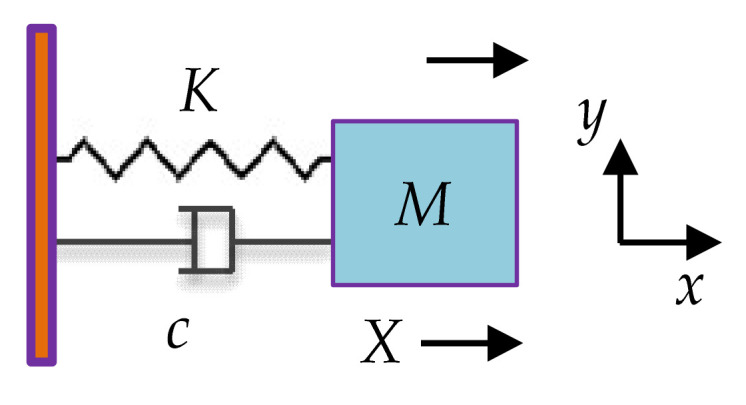
Mass-spring-damper model of a single resonator.

**Figure 7 sensors-22-03857-f007:**
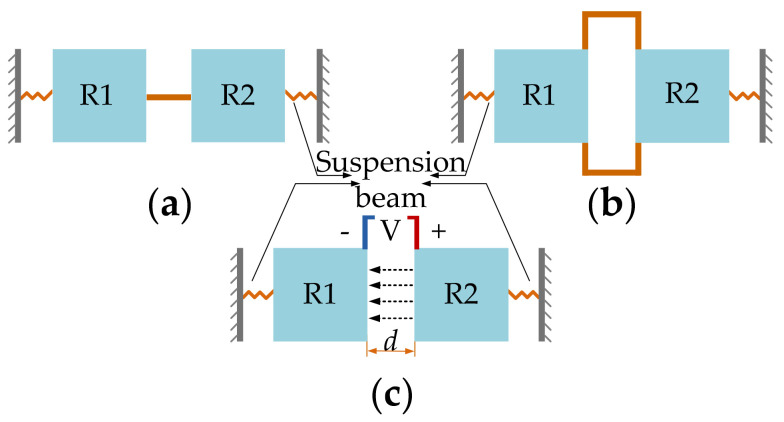
2-DoF coupled BAW resonators with different coupling methods. (**a**) Fixed-fixed straight coupling beam; (**b**) Folded coupling beam; (**c**) Electrostatic coupling. Note: *d* is the lateral transduction gap between the resonator R1 and resonator R2, and the dotted arrow lines indicate the electric field direction.

**Figure 8 sensors-22-03857-f008:**
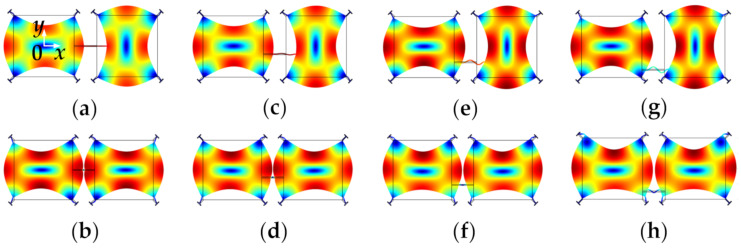
WG modes of a 2-DoF coupled BAW square plate resonator system with the same mechanical coupling beam located at different positions (represented by *y*) simulated in COMSOL. (**a**) Out-of-phase WG mode, *y* = 0 μm; (**b**) In-phase WG mode, *y* = 0 μm; (**c**) Out-of-phase WG mode, *y* = −100 μm; (**d**) In-phase WG mode, *y* = −100 μm; (**e**) Out-of-phase WG mode, *y* = −200 μm; (**f**) In-phase WG mode, *y* = −200 μm; (**g**) Out-of-phase WG mode, *y* = −300 μm; (**h**) In-phase WG mode, *y* = −300 μm. Note: *y* represents the *y*-axis value of the center location of the coupling beam, and the origin is at the center of the first square-plate proof mass. Note: There is a coordinate system in (**a**) with the original point set at the center of the left resonator. The *y*-axis value shows the position of the coupling beam. When the coupling beam is in the center position of both two resonators, *y* = 0. Then the beam moves down to the edge part, the *y* axis value is −100 μm, −200 μm, and −300 μm, respectively.

**Figure 9 sensors-22-03857-f009:**
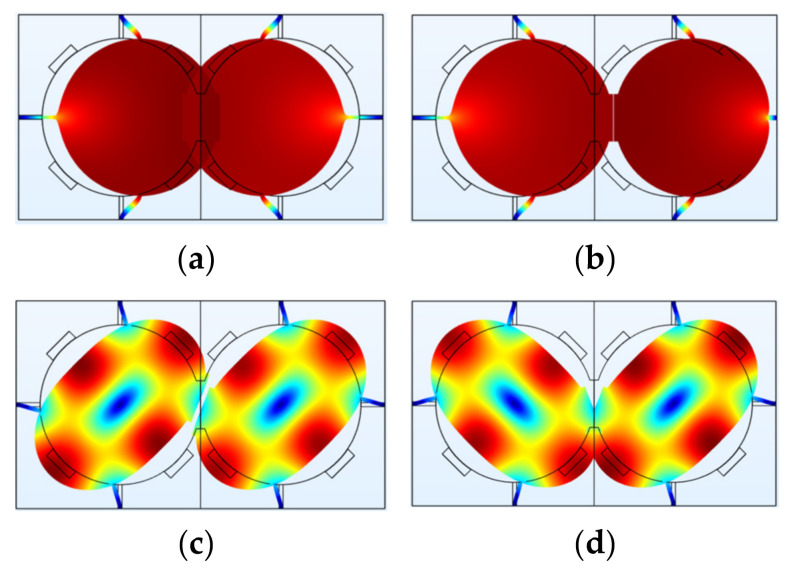
A 2-DoF electrostatically coupled BAW disk resonator system operating in different modes simulated in COMSOL. (**a**) Out-of-phase mode 1; (**b**) In-phase mode 1; (**c**) Out-of-phase mode 2 (WG mode); (**d**) In-phase mode 2 (WG mode). The deformation of resonators is magnified. In reality, the frequencies of the two modes are prone to be aliased due to the limited Q factor.

**Figure 10 sensors-22-03857-f010:**
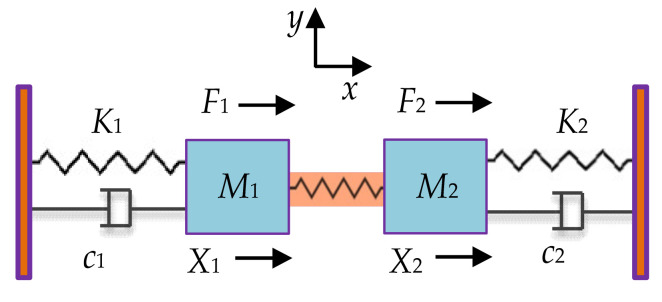
The lumped parameter model of a 2-DoF resonator.

**Figure 11 sensors-22-03857-f011:**
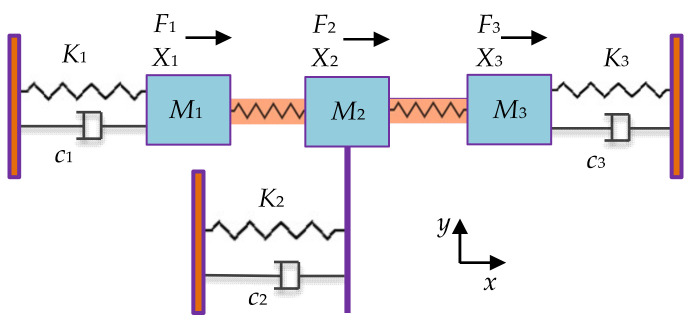
The lumped parameter model of a 3-DoF resonator.

**Figure 12 sensors-22-03857-f012:**
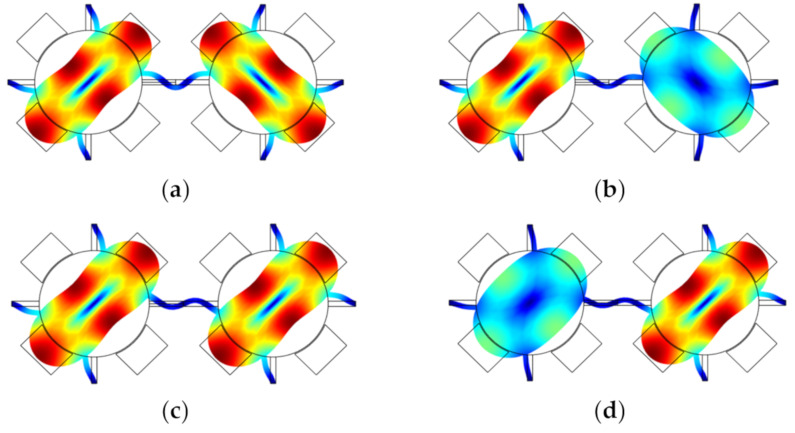
A 2-DoF weakly coupled BAW disk resonator system with mode localization simulated in COMSOL. (**a**) Out-of-phase mode without perturbation; (**b**) Out-of-phase mode with a small mass perturbation on the right resonator; (**c**) In-phase mode without perturbation; (**d**) In-phase mode with a small mass perturbation on the right resonator.

**Figure 13 sensors-22-03857-f013:**
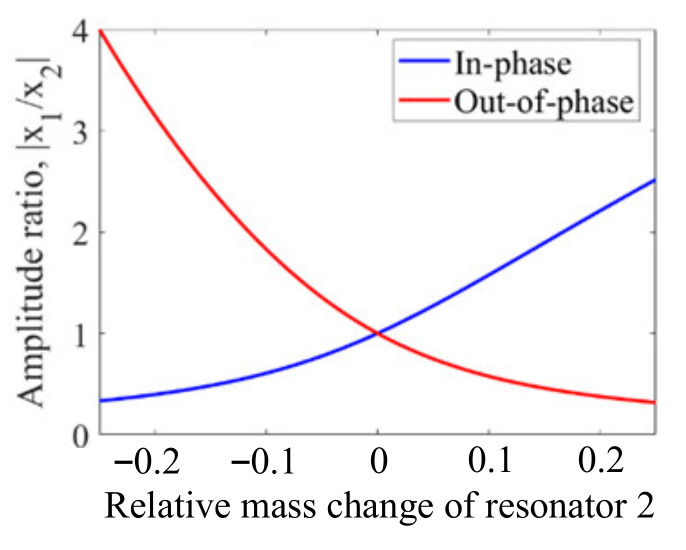
Relationship between the amplitude ratio change and the relative mass change on resonator 2.

**Figure 14 sensors-22-03857-f014:**
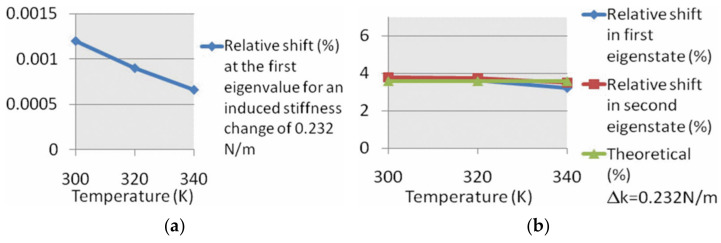
Effect of temperature change on the normalized sensitivity of resonators(Reprinted with permission from Ref. [83]). (**a**) 1-DoF resonators; (**b**) 2-DoF resonators.

**Figure 15 sensors-22-03857-f015:**
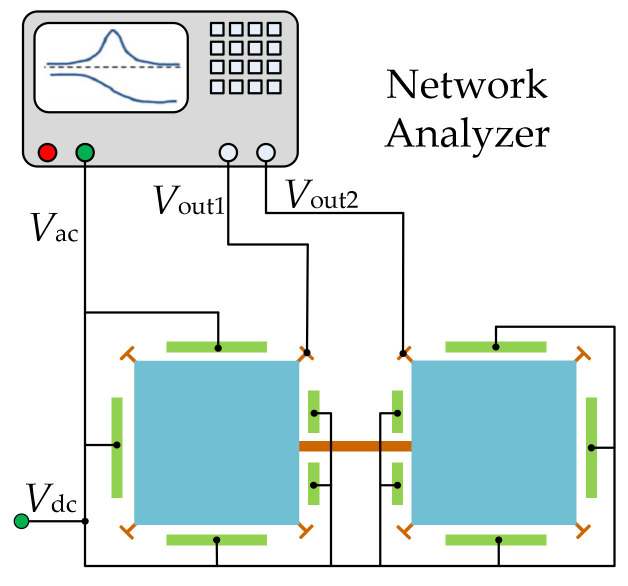
Schematic of a one-port configuration for a 2-DoF square-plate BAW resonator.

**Figure 16 sensors-22-03857-f016:**
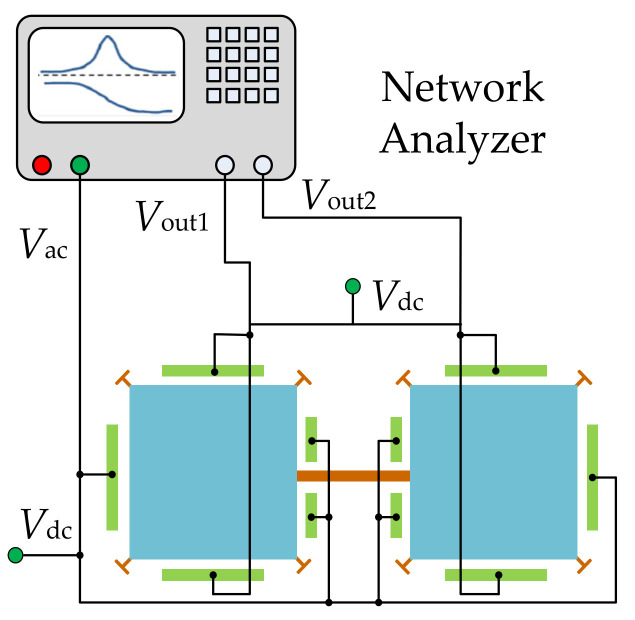
Schematic of a two-port configuration for a 2-DoF square-plate resonator system.

**Figure 17 sensors-22-03857-f017:**
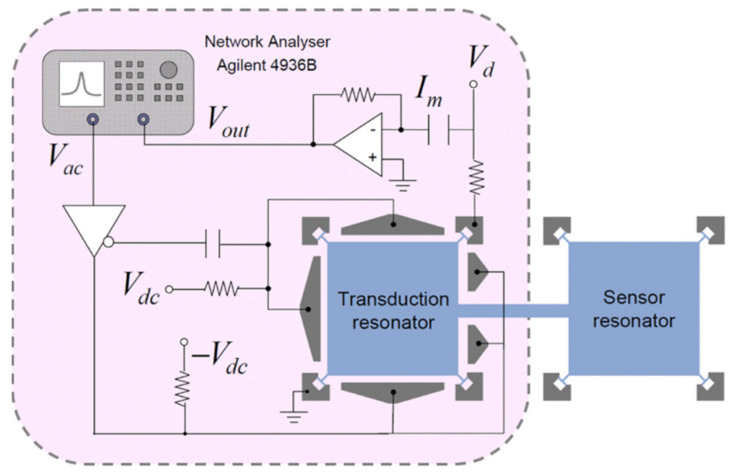
Measurement setup for a coupled BAW resonator system using capacitive actuation and piezoresistive sensing (Reprinted with permission from Ref. [40]).

**Figure 18 sensors-22-03857-f018:**
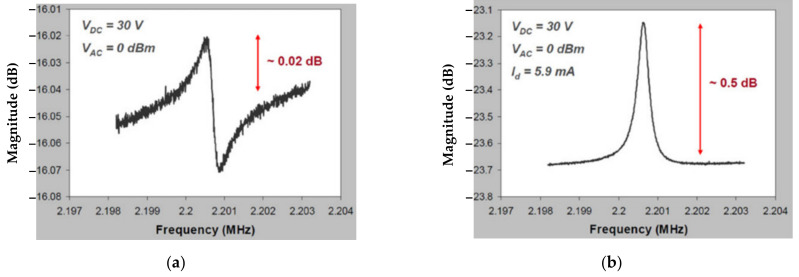
The frequency response measured for a BAW resonator with square extensional (SE) mode in the air (Adapted with permission from Ref. [95]). (**a**) One-port capacitive actuation and sensing method; (**b**) One-port capacitive actuation and piezoresistive sensing method.

**Figure 19 sensors-22-03857-f019:**
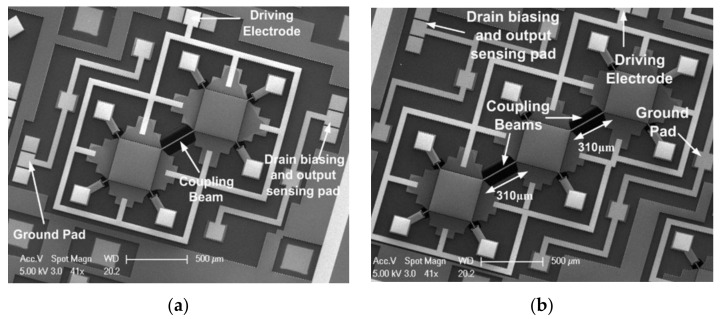
SEM image of the fabricated BAW rectangular-plate resonators with capacitive actuation and piezoresistive sensing setup (Adapted with permission from Ref. [37]). (**a**) A 2-DoF coupled BAW resonator device; (**b**) A 3-DoF coupled BAW resonator device.

**Figure 20 sensors-22-03857-f020:**
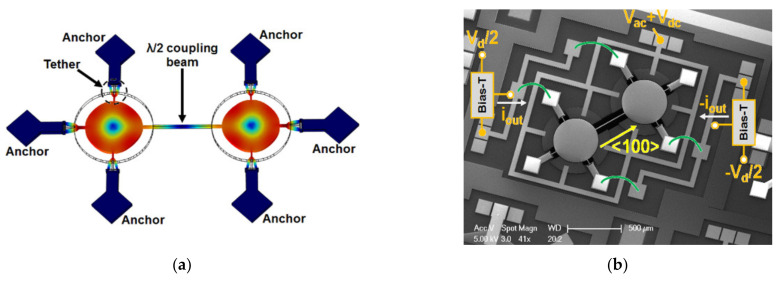
(**a**) FEM simulation of 2-DoF BAW disk resonators with in-phase extensional mode shown; (**b**) SEM image of the fabricated 2-DoF BAW disk resonators with capacitive actuation and piezoresistive sensing. Reprinted with permission from Ref. [38].

**Figure 21 sensors-22-03857-f021:**
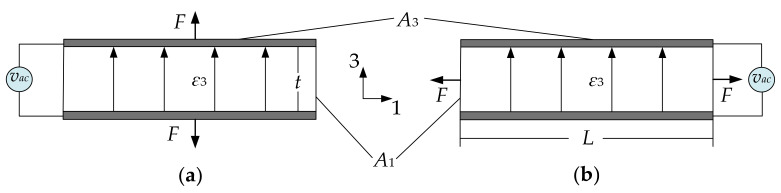
Piezoelectric transduction configuration. (**a**) Longitudinal configuration; (**b**) Transverse configuration. Note: *t* is the thickness of the piezoelectric layer, *A*_1_ is the cross-sectional area of the piezoelectric layer, *A*_3_ is the electrode area, *ε*_3_ is the intensity of the electric field, and *F* is the generated force.

**Figure 22 sensors-22-03857-f022:**
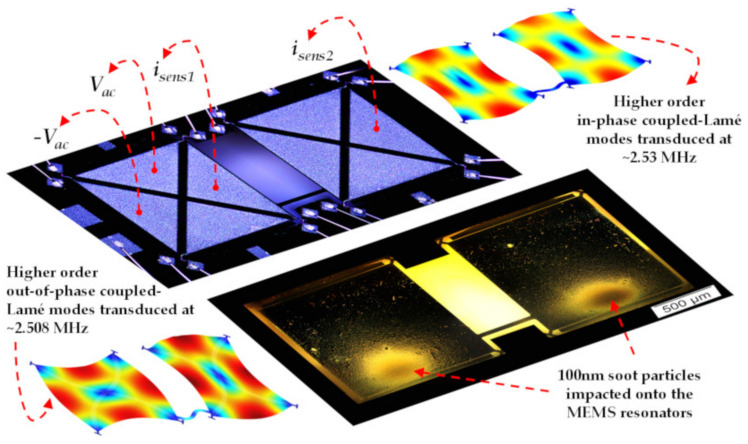
Schematic of the piezoelectric transduction mechanism applied to a 2-DoF weakly coupled BAW square-plate resonator device vibrating at in-phase and out-of-phase WG modes. The mass perturbation is conducted by applying particles onto the bottom surface of the resonators. Reprinted with permission from Ref. [42].

**Figure 23 sensors-22-03857-f023:**
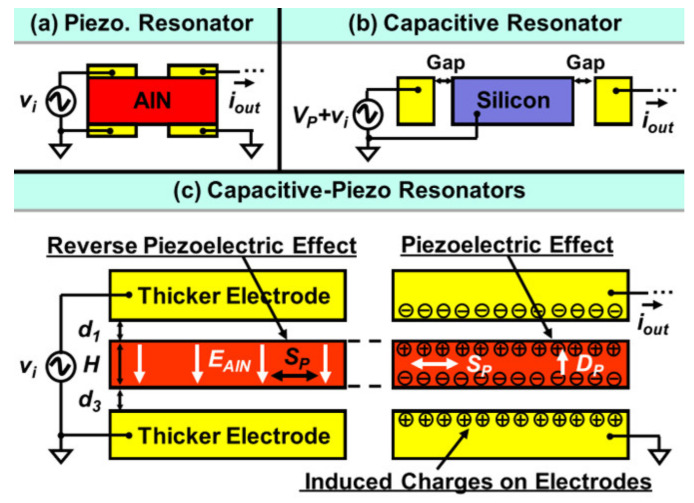
Working principle of a conventional piezoelectric resonator, capacitive resonator and novel capacitive-piezo resonator (Reprinted with permission from Ref. [114]).

**Figure 24 sensors-22-03857-f024:**
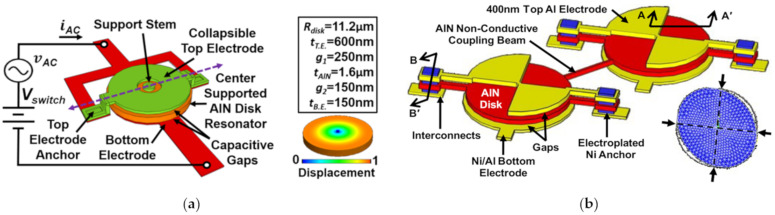
(**a**) Schematic of a 1-DoF capacitive-piezo disk resonator operated at the radial contour mode with the anchor at the center (Reprinted with permission from Ref. [115]); (**b**) Schematic of a 2-DoF capacitive-piezo disk resonator device operated at the WG mode with the anchor connected to the resonator body by a suspension beam (Reprinted with permission from Ref. [114]).

**Figure 25 sensors-22-03857-f025:**
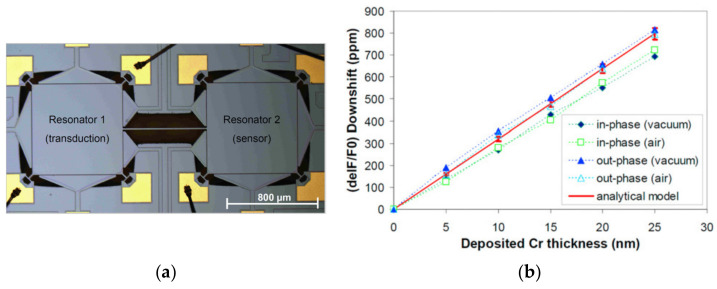
(**a**) Optical micrograph of the 2-DoF BAW resonator system; (**b**) Comparison between the analytical frequency shift and the measured one with respect to the deposited Cr film thickness. Reprinted with permission from Ref. [39].

**Figure 26 sensors-22-03857-f026:**
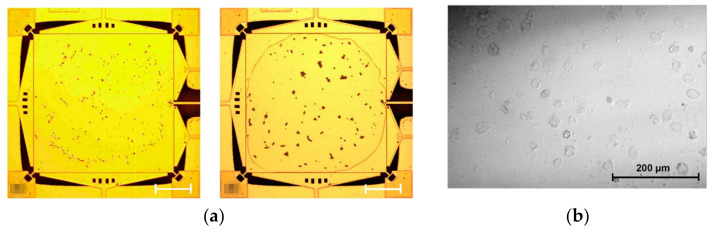
Optical micrograph of attached analytes. (**a**) SCPMs with diameters of 5.61 μm and 15.68 μm; (**b**) High Five insect cells. Reprinted with permission from Ref. [40].

**Figure 27 sensors-22-03857-f027:**
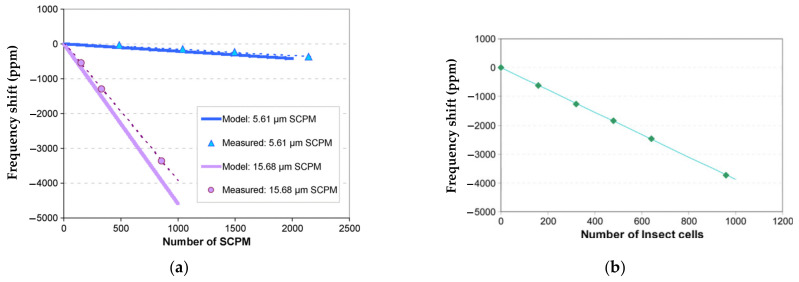
Relationship between the frequency shift and the number of analytes attached to the resonator from. (**a**) SCPMs; (**b**) High Five insect cells. Reprinted with permission from Ref. [40].

**Figure 28 sensors-22-03857-f028:**
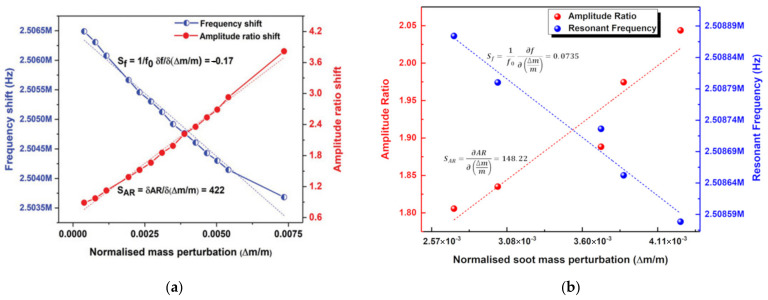
(**a**) Normalized sensitivity based on frequency shift and amplitude ratio shift (Reprinted with permission from Ref. [41]); (**b**) Normalized sensitivity based on frequency shift and amplitude ratio shift from (Reprinted with permission from Ref. [42]).

**Figure 29 sensors-22-03857-f029:**
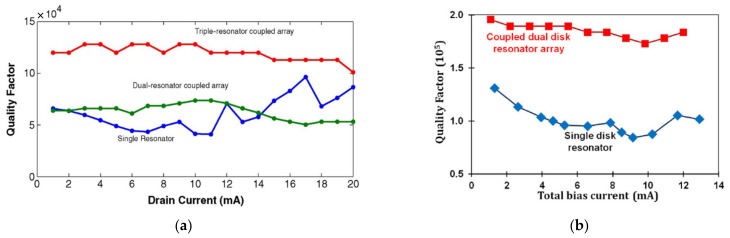
(**a**) Comparison of Q factors among the single, 2-DoF and 3-DoF square-plate BAW resonators as the drain current (Reprinted with permission from Ref. [37]); (**b**) Comparison of Q factors between the single and 2-DoF disk BAW resonators with bias current increasing (Reprinted with permission from Ref. [38]).

**Figure 30 sensors-22-03857-f030:**
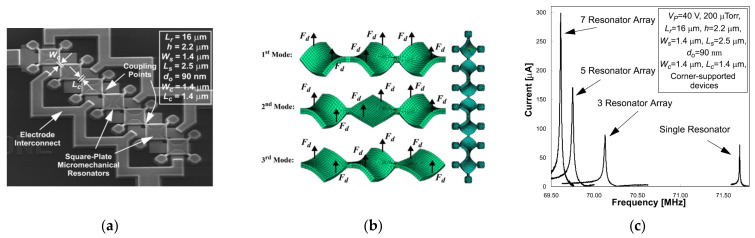
(**a**) SEM image of a 7-DoF strongly coupled BAW resonator system with anchors at the center; (**b**) FEM simulated out-of-plane motion of the coupled resonator systems; (**c**) Frequency response of several multi-DoF coupled resonators. Reprinted with permission from Ref. [37].

**Figure 31 sensors-22-03857-f031:**
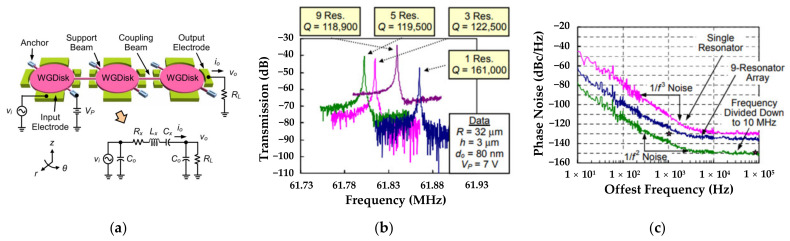
(**a**) Schematic of a 3-DoF strongly coupled disk resonator system with the equivalent electric circuit; (**b**) Measured frequency response for several multi-DoF disk resonator systems; (**c**) Relationship between the offset frequency and phase noise. Adapted with permission from Ref. [118].

**Figure 32 sensors-22-03857-f032:**
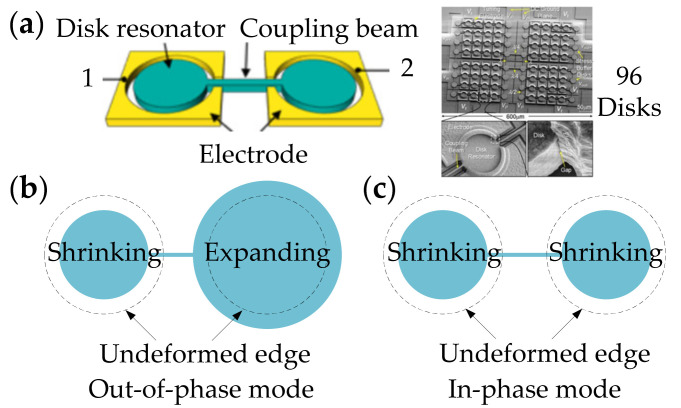
(**a**) Schematic and SEM image of a 2-DoF and 96-DoF coupled BAW disk resonator filter system; (**b**) Revised schematic of the out-of-phase radial contour mode shape; (**c**) Revised schematic of the in-phase radial contour mode shape. Adapted with permission from Ref. [80].

**Figure 33 sensors-22-03857-f033:**
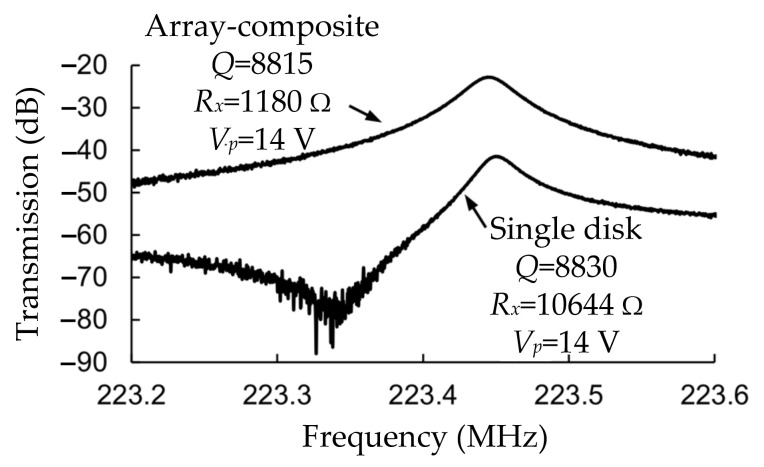
A comparison of frequency spectra between a 30-DoF disk system and a single disk system. Adapted with permission from Ref. [80].

**Table 1 sensors-22-03857-t001:** Description of several classic BAWs.

Type of Waves	Characteristics	Schematic of Wave Propagation
Longitudinal wave	Propagation through the medium in the same or opposite direction of particle vibration or oscillation. It is also called pressure waves.	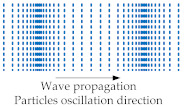 Adapted with permission from Ref. [25].
Transverse wave	Propagation through the medium in the direction perpendicular to particle oscillation direction. It is also called shear-mode waves.	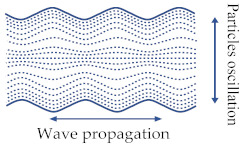 Adapted with permission from Ref. [4].
Lamb wave	It exists in thin plates, also called the plate wave, and the particle motion lies in the plane that is perpendicular to the plate and the propagation direction of the wave [25]. It is divided into symmetric waves and antisymmetric waves [24].	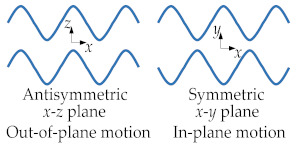 Adapted with permission from Ref. [24].

**Table 2 sensors-22-03857-t002:** Dimensions and materials of a 2-DoF coupled BAW square plate resonator system modeled in COMSOL (Adapted with permission from Ref. [26]).

Parameters	Value
Length of the square plate (μm)	800
Thickness of the device (μm)	25
Length and width of the stem (μm)	67.2 and 14.1
Length and width of the cap (μm)	70.7 and 10.6
Length and width of the coupling beam (μm)	400 and 14
Materials selection for FEM simulation	Single crystal silicon (SCS)

**Table 3 sensors-22-03857-t003:** The in-phase and out-of-phase mode resonant frequencies of a 2-DoF coupled BAW square plate resonator system with the same mechanical coupling beam located at different positions obtained by FEM simulation in COMSOL.

Location of Coupling Beam	In-Phase Mode (Hz)	Out-of-Phase Mode (Hz)	Frequency Difference
*y* = 0 μm	4,702,389	4,769,160.74	1.42%
*y* = −100 μm	4,704,931.73	4,762,091	1.21%
*y* = −200 μm	4,711,448.66	4,746,444.54	0.74%
*y* = −300 μm	4,717,826.99	4,730,406.99	0.27%

**Table 4 sensors-22-03857-t004:** Dimensions and materials of a 2-DoF electrostatically coupled BAW square resonator system modeled in COMSOL.

Parameters	Value
Diameter of the disk plate (μm)	1500
Thickness of the device (μm)	30
Lateral transduction gap (μm)	2
Length and width of the stem (μm)	225 and 37.5
Length and width of the convex plate (μm)	5.5 and 450
Materials selection for FEM simulation	Single crystal silicon (SCS)

**Table 5 sensors-22-03857-t005:** Differences in resonant frequencies of a 2-DoF electrostatically coupled BAW disk resonator system operated at different vibrational modes obtained by FEM simulation in COMSOL.

Vibration Modes	Out-of-Phase Mode (Hz)	In-Phase Mode (Hz)	Frequency Difference
Mode 1	365,455.475	365,537.439	0.02%
Mode 2 (WG mode)	2,654,149.09	2,654,150.75	0.00006%

**Table 6 sensors-22-03857-t006:** Differences in resonant frequencies of a 2-DoF mechanically coupled BAW square resonator system operated at different vibrational modes obtained by FEM simulation in COMSOL.

Vibrational Modes	In-Phase Mode (Hz)	Out-of-Phase Mode (Hz)	Frequency Difference
Mode 1	381,093.019	411,584.939	8%
Mode 2 (WG mode)	4,711,448.66	4,746,444.54	0.74%

**Table 7 sensors-22-03857-t007:** Dimensions and materials of a 2-DoF weakly coupled BAW disk resonator system modeled in COMSOL.

Parameters	Value
Diameter of the disk plate (μm)	200
Thickness of the device (μm)	30
Lateral transduction gap (μm)	2
Length and width of the stem (μm)	50 and 10
Length and width of coupling beams (μm)	125 and 10
Materials selection for FEM simulation	Single crystal silicon (SCS)

**Table 9 sensors-22-03857-t009:** Performance comparison of the reported 2-DoF coupled BAW MEMS resonators as mass sensors.

References	[39]	[40]	[41,42]
Vibration mode	Extensional mode	Extensional mode	WG mode
Actuation way	Capacitive	Capacitive	Piezoelectric
Sensing way	Piezoresistive	Piezoresistive	Piezoelectric
Frequency (MHz)	5.492, 5.423	5.492; 3.145	~2.5
Q factors	5139, 8505	5139; 8676	1773.8
Mass perturbation	Cr film	Microbeads and cells	Particles
Characterize sensitivity	Frequency shift	Frequency shift	Frequency shift; AR shift
Sensitivity	34 Hz/ng	−12 Hz/cell	0.0735; 148.22
Detection limit	Not mentioned	1.46/36 ng/cm^2^; ~1.68 dried cells	Not mentioned

## Data Availability

Not applicable.

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
