# Peer review of "A Review on Coupled Bulk Acoustic Wave MEMS Resonators"

_sensors, 2022, doi:10.3390/s22103857_

Round 1

Reviewer 1 Report

These minor modifications should be implemented before the publication of this work.

Line 118: What is the software FEM used for the representation of Figures 1, 2, 3, 8, 9?

The results presented in Table 2, 3, 4 come from simulation?

Some reference for equation 10, 11?

The representation of Fig. 14 should be improved.

Fig. 18: Are the results measured or simulated? Fig 18.b appears with letter “e” in the figure.

Some reference should be introduced for Figure 23(a) and (b)

The figure caption of Figure 28 is in color red.

Author Response

Dear

Please see the attachment. Thanks for your comments.

Best regards,

Linlin Wang

Reviewer 2 Report

The manuscript, "A review on coupled bulk acoustic wave MEMS resonators", is a review paper discussing the coupled BAW MEMS resonantor theoretical analysis and practical applications. The manuscript provides a good coverage of BAW resonantor thoery, characterization methods, and transduction methods. The weakness of the manuscript relies on the practical applications. The references and examples are limited and the outdated. Thus, a major revision is resommneded and below are my detailed comments:

  1. There are some grammar issues in the manuscript. For example, line 26, "Micro-electro-mechanical system (MEMS) is a research field that use mechanical structures often..." "use" requires an "s". Line 29-30, "A sub-class that has recently received increasing attention and has become one of the most important building blocks of MEMS devices, are micromachined resonant devices", "has" and "are micromachined..." are controdictory. Please pay attention to the plural form. Similarly for Line 33-34. Please perform a proof read.
  2. The references are out-dated. The latest ones are from 2014 and 2016. Please add more recent publications. 
  3. From Section 4, the coupled BAW device examples include sensor, oscillators, and filters. However, in section 2, only sensors key metrics and characterization methods are discussed. Please add similar discussions for oscillators and filters.
  4. The images are blur and the tables have poor formats. Please improve.
  5. The author demonstrates schematics and simulation modal analysis results, from Fig. 1 - 12. Are these results from literatures? If so, please add the references in the figure caption. If the results are from authors' own simulation results, please add devices dimension details, including the gemoetry, thickness, material properties, anchor locations, etc. It is confusing to discuss about device performance without the details. For example, in Table 3&4, how are the resonance frequencies calculated? In Fig. 4, how the location of the coupling beam y is defined and simulated?
  6. Section 2.2.2 discussed about the 2-DOF resonantor. Section 2.2.3 and 2.2.4 discussed about weak coupling of 2-DOF and 3-DOF system. Did 2.2.2 discussed strong coupling? Please add more organized discussion about coupling type and 2-DOF/3-DOF system or multi-DOF system. And why no simulation results are shown for section 2.2.3 and 2.2.4? What's the criteria to differentiate strong and weak coupling and what's the pros and cons in terms of each system design?
  7. I am not too clear about Fig. 4 different suspension system. Could you elaborate more about Fig. 4 instead of referring to ref 26 and 72? The figure is not too clear how the strutures are designed.
  8. Fig. 5, what's the pros and cons for different anchoring design at different positions?
  9. Please explian d in Fig. 7, as well as the dotted arrow lines.
  10. Table 5 is a very rough table and it is confusing. First of all, why actuation for and sensed motion current is discussed in this table for comparison? Second, the author left blank on the piezoresistive method for force acutation. Why it is like that? Third, in the pros of piezoelectric methods, the author mentioned "No DC required". Please further elaborate on this point. Then in the cons, the author mentioned piezoresistive "high power consumption". Please further elaborate on this. It is recommended to give detailed numbers for all the three methods. Meanwhile, electrastaic has limitations in fluidic environment. Why piezoresistive and piezoelectric methods can be applied in fluidic environment? Please provide references. 

Author Response

(The authors gave the same response as above.)

Reviewer 3 Report

Most of the manuscript were popular science which was worthless to researchers in related feild. Simulations and schemes were use too much. Experimental data or published results should be analyzed and discussed in review paper. For BAW resonators, what were the highest frequency and quality factor? Which structures and materials were utilized? In practical applications, how structures and materials affected the performance? Which group were carrying on this study? Which issues should be paid attention to? Scientific imformations were lack in this manuscript.

Author Response

(The authors gave the same response as above.)

Round 2

Reviewer 2 Report

The author has addressed all my comments. 

Reviewer 3 Report

The level of this manuscript is unsatisfied with this journal. Those hot topics or key issues or cutting edges are NOT summarized. The significance of this manuscript is lack.